# Neuronal correlates of endogenous selective attention in the endbrain of crows
Lukas Alexander Hahn [1,2] ✉, Erica Fongaro[1,2] & Jonas Rose [1] ✉

The ability to direct attention and select important information is a cornerstone of adaptive behavior. Directed attention supports adaptive cognitive operations underlying flexible behavior, for example in extinction learning, and was demonstrated behaviorally in both mammals and in birds. The neural foundation of such endogenous attention, however, has been thoroughly investigated only in mammals and is still poorly understood in birds. And despite the similarities at the behavioral level, cognition of birds and mammals evolved in parallel for over 300 million years, resulting in different architectures of the endbrain, most notably the absence of cortical layering in birds. We recorded neuronal signals from the nidopallium caudolaterale, the avian equivalent to mammalian pre-frontal cortex, while crows employed endogenous attention to perform change detection in a working memory task. The neuronal activity profile clearly reflected attentional enhancement of information maintained by working memory. Our results show that top-down endogenous attention is possible without the layered configuration of the mammalian cortex.

Endogenous attention is a cornerstone of complex cognition such as awareness and (sensory) consciousness[1]. By focusing neural resources on relevant information, attention can optimize limited cognitive resources such as working memory (WM), the maintenance and manipulation of information over short periods of time[2]. Behavioral correlates of attention processes are observed throughout the animal kingdom[3–5]. Two forms of attention of different complexities can be distinguished[6,7]. Exogenous attention is considered evolutionary primitive, automatic, involuntary, and transient, and it is caused by a stimulus in the environment that "hijacks" mental resources[4]. This form of attention has been observed in many animals and relies on relatively simple neuronal circuitry in a bottom-up fashion[8]. The other form is endogenous attention, which is considered voluntary, sustained, and goal-directed. It is actively engaged in a top-down fashion by cognitive-control circuitry that is evolutionary derived and likely evolved convergently across different animal groups[4,9]. Evidence for endogenous attention has been observed in several mammals, namely primates, rats, and mice[10–17] and in birds, e.g., crows[1,18,19], owls[20–22], chicken[23], and peafowl[24]. Overall, although the avian brain lacks the mammalian neocortex, both primates and corvids have also independently evolved comparable forms of complex cognition and behavior[25,26]. How do mammals and birds, whose brains have evolved independently for about 320 million years[27], end up showing such similar cognitive feats? This makes a comparison of these animals particularly intriguing.

Attentional processes are essential to stimulus selection in WM and vital for goal-directed behavior, as they allow to safeguard relevant information from loss when capacity limitations are reached[28]. Thus, the involvement of telencephalic regions in attention and selection has been investigated in detail in mammals[3]. In primates, endogenous attention and WM have been investigated as executive functions that rely on overlapping neural circuits centered around the prefrontal cortex[9,29–31]. In contrast, in birds, attentional processing has been studied at the neuronal level for the evolutionarily conserved midbrain stimulus selection network, which critically includes the optic tectum of birds (homologous to the superior colliculus in mammals)[8,32]. However, descriptions of attentional correlates in the telencephalon of birds are sparse. Only the telencephalic arcopallial gaze field has been investigated for attentional processing through its connections to the midbrain selection network[33,34].

We used a delayed change detection working memory task that required attention to spatially presented color information to record neuronal activity from the nidopallium caudolaterale (NCL) of two carrion crows (*Corvus corone corone*). Despite its lack of neocortical organization, the NCL, a higher associative region of the telencephalon of birds, serves similar functions to the mammalian prefrontal cortex (PFC)[25,26,35–37]. Neuronal correlates of WM in the NCL of carrion crows are comparable to primate PFC, in the spatial domain[38–40], for numerosity[41–43], abstract rules[44,45], capacity limitations[46], and oscillatory dynamics[47].

[1]Neural Basis of Learning, Institute of Cognitive Neuroscience, Faculty of Psychology, Ruhr University Bochum, Bochum, Germany. [2]These authors contributed equally: Lukas Alexander Hahn, Erica Fongaro. ✉e-mail: lukas.hahn@ruhr-uni-bochum.de; jonas.rose@ruhr-uni-bochum.de

In two versions of our paradigm, we either cued a relevant location to guide attention or did not present attention cues. We expected to observe volitional (endogenous) attention directed towards the cued location in pre-cue trials. In contrast, birds were expected to split working memory resources between the three locations in no-cue trials. By comparing neuronal activity between pre-cue trials and no-cue trials, we reasoned to be able to identify the attentional signal interacting with the information about the targeted color. However, we found that in no-cue trials, the birds made use of implicit strategies, resulting in endogenous attentional selection in the same way as in pre-cue trials. While unexpected, this allowed us to directly compare neuronal signals between conditions when a stimulus was attended following the presence of the pre-cue or when it was attended following a purely internal decision to attend towards a specific stimulus location. By comparing neuronal activity of the different trial types, we were able to identify that the neuronal correlate of attention we recorded in pre-cue trials was effectively the same as the one that followed a purely endogenous internal strategy.

## Results

Two carrion crows performed a visual change detection task. In this task, the birds were required to determine if, following a memory delay, one out of three colored squares had changed its color (change-trial) or if all colors remained the same (no-change-trial, Fig. 1A, see "Methods" for details). We analyzed two different trial types. In pre-cue trials, a visual cue indicated the only relevant location where a change could occur, directing the birds' attention toward the color at that specific location. In no-cue trials, the same task was performed without the cue, and the birds were free to either attend all three locations simultaneously or selectively attend a location of their own choosing. The birds performed very well in pre-cue trials and focused on single locations in the no-cue trials. We recorded neuronal activity from the NCL of the behaving crows and analyzed the resulting single-cell data, testing the hypothesis that attention would increase information processing related to color. Due to the relative complexity of the behavioral and neuronal results in the no-cue trials, we will first report on the results of the pre-cue trials, showing the effect of endogenous attention directed towards a relevant stimulus based on an external cue. We will subsequently illustrate how these results are reflective of endogenous attention by comparing them to the results obtained from no-cue trials, where the animals also made use of endogenous attention.

### Crows excel in change detection at cued locations

Both birds exhibited very high performance in the change detection task in pre-cue trials, reflected in the very high hit rates (median (Mdn) performance in percent: 98.08, 98.25, 98.11 for bird 1; and 100, 89.46, 89.07 for bird 2, for top middle and bottom location, respectively) and correct rejection rates (Mdn in percent: 97.30, 96.61, 94.29 for bird 1; and 100, 97.25, 91.51 for bird 2, for top middle and bottom location, respectively, Fig. 1B, C). For bird 1, this performance was independent of location for hit rates ($X^2(2,84) = 2.59$, $p = 0.27$) and correct rejections ($X^2(2,84) = 0.76$, $p = 0.68$). In contrast, it differed significantly between locations for bird 2 for hit ($X^2(2,33) = 6.25$, $p = 0.0439$), and for correct rejections ($X^2(2,33) = 7.84$, $p = 0.0198$). Post hoc analysis confirmed that hit rates and correct rejection rates for the top location were significantly higher than for the bottom location ($p < 0.05$, Fig. 1C), but there were no significant differences between top and middle and between middle and bottom location (both $p > 0.05$). If the birds reported a change of colors, they virtually always pecked on the location where the change had happened.

### Neuronal activity in NCL

We recorded a total of 356 neurons (bird 1: 163, bird 2: 193 neurons) from crow NCL (Fig. 2A, nomenclature following the description of von Eugen et al. (2020)[48]). Among these neurons, 56% showed significantly different firing rates between different conditions. We rationalized that if the NCL were involved in attention, it should represent (a) information about the cue location, and (b) in turn, attention should result in an improved

representation of color at a cued location. Therefore, we first tested if neuronal activity in pre-cue trials was different based on the factors: location (presence of the pre-cue at either the top, middle, or bottom location), and color (color 1 or color 2 at any location, irrespective of the other colors on screen), or both. We found that 29% of all neurons showed exclusive significance for the factor location, while only 4% of neurons had exclusive significance for color, and 23% of neurons showed significant modulation of firing rates for both factors (Fig. 2B, all significances based on Bonferroni adjusted $p$-values for multiple comparisons, see methods for details, see Fig. S1A, for individual birds).

### Cue location was the dominant factor driving neuronal activity

Firing rate differences between the cue locations appeared while the cue was visible (cue period), and firing rate differences often remained throughout the rest of the trial. For example, the neuron shown in Fig. 2C significantly increased firing rate in the cue phase if a cue was present at any location, in comparison to no-cue trials (black trace, peak difference at the end of the pre-cue phase, 140 ms after cue onset, $F(3427) = 89.64$, $p < 0.0001$, $\omega^2 = 0.3832$; post hoc comparisons of pre-cue locations against no-cue all $p < 0.0001$). Firing rate was highest if the cue appeared at the middle location (blue trace) and was significantly lower for both the top (yellow trace) and bottom (red trace) cue location (peak difference at the end of the cue phase, 140 ms after cue onset, post hoc comparisons of pre-cue locations against each other all $p < 0.0001$). Throughout the trial the neuron continued to intermittently increase firing rate peaking again in the early delay (200 ms after sample-offset, $F(3427) = 25.75$, $p < 0.0001$, $\omega^2 = 0.1478$) in trials with a cue at the middle and bottom location and during the compare phase (300 ms after comparison onset, $F(3427) = 24.74$, $p < 0.0001$, $\omega^2 = 0.1427$) in trials with the cue at the middle location. Note that the pre-cue was only visible in the cue phase at the start of the trial, thus, firing rate differences between cue locations later in the trial were not directly driven by visual stimulation. Other example neurons showed different spatial preferences (e.g., to the top and bottom position, Fig. S1B, C, respectively) or different neuronal activity exclusively during the pre-cue phase.

We quantified information about cue location by calculating the effect size ($\omega^2$) for the factor location (three levels) in a one-way ANOVA and interpreted it as the percent explained variance (PEV, the amount of variance explained by the cue location). The neuronal population of NCL encoded information about the cue location throughout the entire length of the trial (Fig. 2D, all recorded neurons, pooled across animals, for individual animals see Fig. S1D). The proportion of significant neurons in the population, at any given time point, remained stable at around 10% (Fig. 2D, purple line). Information peaked during the cue phase (when the cue was visible on screen), again during the sample phase (when the color at the cue location had to be memorized), and finally, during the compare phase (when the color at the cue location had to be compared to the memorized color). We tested if the amount of information about the cued location was sufficiently large to allow decoding of the relevant location, using a neuronal classifier. We trained the classifier (support-vector machine, see methods for details) with the neuronal activity of all recorded neurons per bird that had significant information about the pre-cue location (i.e., "cue location" and "cue location + color" groups in Fig. 2B). We cross-validated the performance of the classifier to correctly label a given test-trial (that belonged to any of the three possible cue locations) by training on a subset of all cue trials (80%) and testing on the remaining cue trials (20%). We determined the significance of the classification of pre-cue location by randomized permutation of the labels of the training set (see methods for details). Classification performance peaked when training bin and testing bin were the same or temporally close, indicating that the classifier managed to correctly classify cue location based on the local neuronal pattern. There were also non-adjacent time points that exceeded chance levels at different times throughout the trial, which indicated (limited) stability of information of neuronal firing patterns (Fig. 2E, F, black outline indicates when values exceeded significance threshold at alpha = 5%, also see Fig. S1E). The patterns of decoding

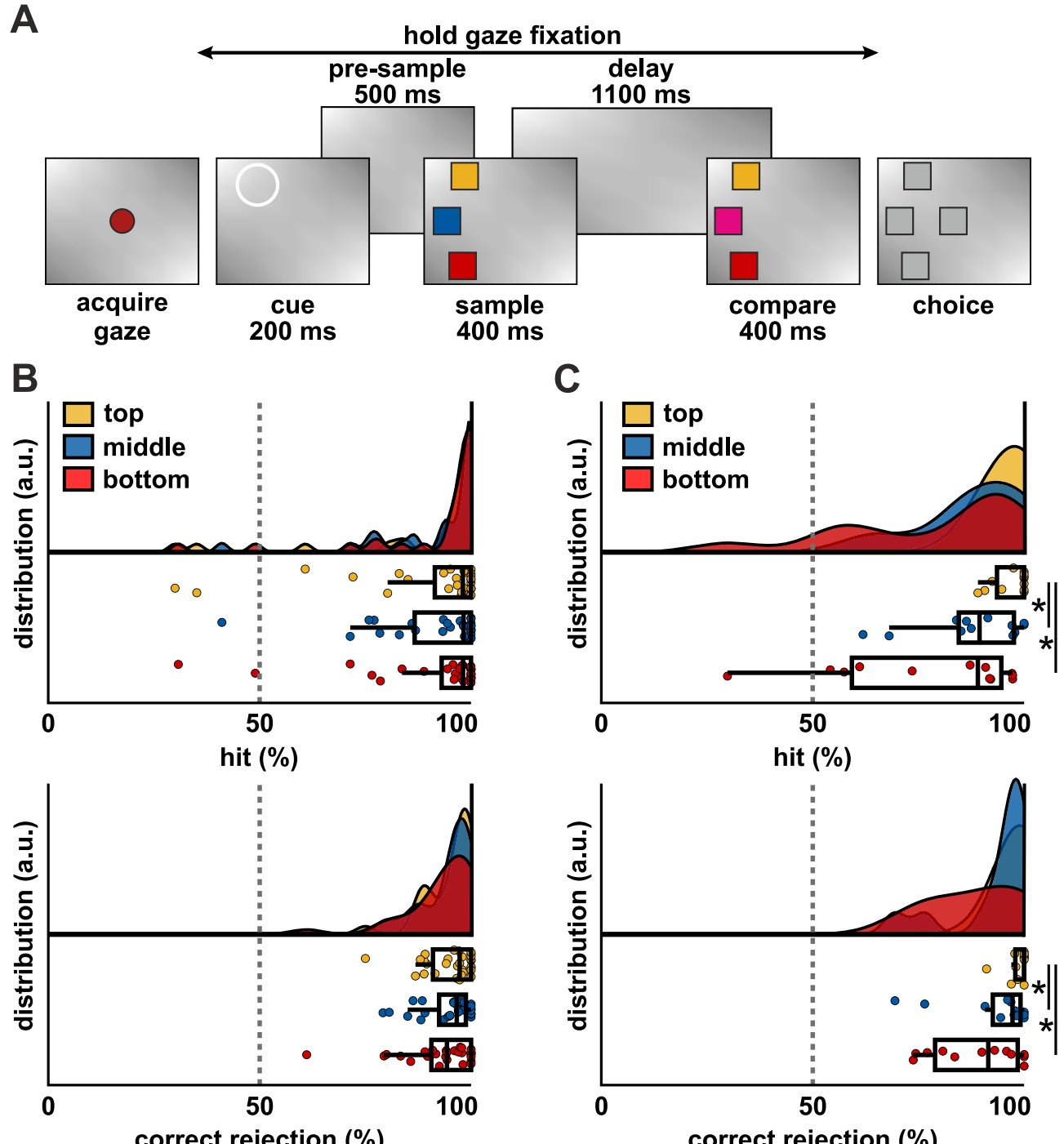

**Fig. 1 | Task and behavioral results. A** The visual change detection task. Three colors were presented and, following a delay, reappeared either with one color exchanged or unchanged (each 50% of trials). During choice, the birds were rewarded for correctly indicating a change with a peck to any of the three lateral locations. Birds indicated no change with a peck to the center location. A pre-cue was presented prior to sample presentation to indicate the relevant location (i.e., top, middle, or bottom). The cue indicated that if a change were to occur, it would be at the cued location. **B** Performance of bird 1. The hit rate (correct detection and localization of color change) and correct rejection rate (correct detection of a lack of color change) are shown for each possible cue location. Upper plots indicate the distribution of values (individual sessions) along the performance. Lower plots show all data points ($n = 28$ sessions); boxes indicate first quartile, median, and third quartile. Whiskers extend to 1.5 times the interquartile range. The dashed line indicates the chance level. **C** Same as in (**B**) for bird 2 ($n = 12$ sessions), asterisks indicate significant differences.

performance differed between birds. Along the primary diagonal (where training and testing bins were congruent), classification performance peaked at 92.03% (standard deviation 4.77) for bird 1 during the early sample phase and at 81.49% (standard deviation 5.09) for bird 2 during the late delay. This indicated that the tested neuronal population contained enough information in its pattern of activity to successfully decode the location of the cue. Bird 1 showed strong location encoding during sample and comparison phase that was temporally stable between the two phases (i.e., neuronal activity in sample decoded activity in compare phase and vice versa, illustrated by the islands of high classification values in Fig. 2E),

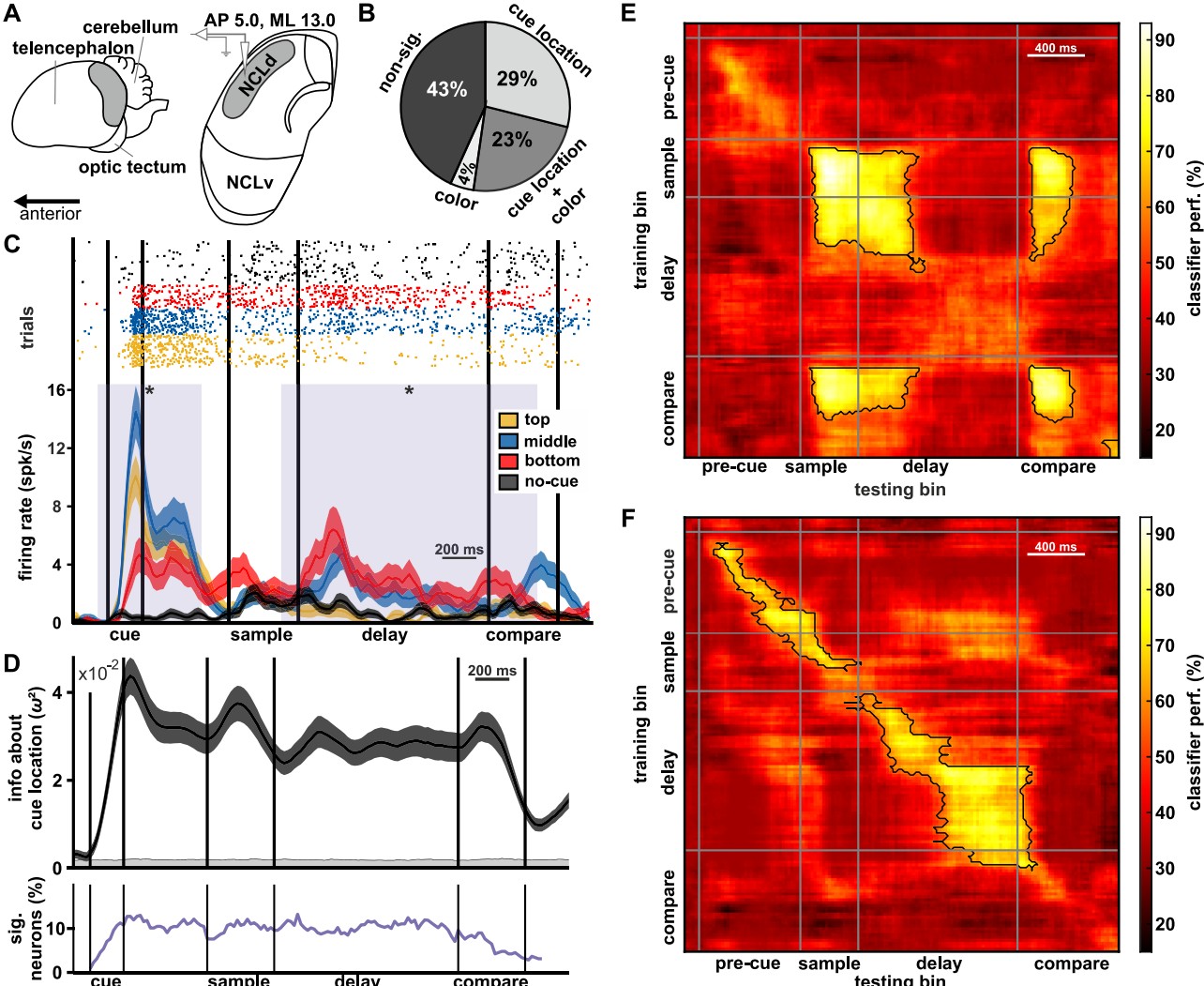

**Fig. 2 | Neuronal activity related to pre-cue location. A** Recording site in the crow brain. Schematic anatomical layout at AP 5.0 NCLd – dorsal nidopallium caudo-laterale, NCLv – ventral nidopallium caudolaterale. Recordings were made throughout the extent of NCLd. **B** Proportions of significant neurons (rounded to the nearest integer) for each test factor ($n = 356$ recorded units). **C** Example neuron showing differential spiking for different cue locations. Upper part: dot-raster plot of neuronal spiking, each dot represents an individual spike, sorted and color coded based on location of the cue (trial numbers $n_{top} = 113$, $n_{mid} = 88$, $n_{bot} = 78$). Trials without a cue ($n_{noCue} = 149$) in black. Lower part: Peri-stimulus time histogram (PSTH) of spike-density functions in different trial conditions. Top, middle, and bottom refer to the location of the cue during the cue phase. The solid line indicates the mean, and the shaded area around the line indicates the standard error of the mean. The purple shading indicates periods when spike-density between the different cue locations was significantly different. Note that due to binning (200 ms bins advanced in 20 ms steps), the rise in firing rate in the PSTH appears to become

significant even before the onset of the cue. Please refer to the dot raster for the precise actual timing of spikes relative to set events. **D** Cue location was encoded throughout the trial by neurons at the population level ($n = 356$ recorded units). In black: mean percent explained variance (PEV, shaded area indicates standard error of the mean) of all neurons from both birds. The gray shaded area indicates PEV of randomized data set. Below in purple: percentage of neurons with significant information at the respective timepoint. **E** Results of an SVM classifier, along the behavioral protocol. For bird 1, cue location was encoded in neuronal activity, mostly during sample presentation and the comparison phase (high classifier performance values, chance level at 33%). *X*-axis and *Y*-axis indicate time bins (200 ms, advanced in 20 ms steps) of the classifier testing and training, respectively. Only neurons with significant activity for the factor location were included in classifier training and testing ($n = 75$ units). Black outlines indicate significant clusters of decoding. **F** Same as in (**E**) for bird 2 ($n = 110$ units). Cue-location during the delay period had higher decoding performance in bird 2.

but only relatively poor decoding performance of cue location prior to the sample onset and during the delay. The lower classifier performance prior to the sample onset, when cue location had already been revealed, might reflect a delay in how fast the animal was able to pick up on the cue location. During the delay, cue location may have been maintained in an activity silent state not observable through firing rate dependent classification. In contrast bird 2 had information about cue location consistently throughout the trial, peaking towards the end of the delay, but only very little temporal consistency between phases (only limited and non-significant performance values across trial phases, Fig. 2F). The animal was fast at encoding the cued location, indicated by the significant

classification of cue location shortly after cue presentation, and relied on a more sustained form of neuronal firing to maintain the cue location across the delay. Beyond the primary training-testing diagonal, classification had varying levels of performance, indicating only limited temporal generalization in both birds. Overall, these results suggest that cue location was maintained at least until the comparison phase, either through reactivation of neuronal activity patterns of the sample phase (bird 1) or in a more continuous and temporally independent fashion (bird 2). In both cases, when the comparison phase came up and the animal had to detect a possible change of color, information about the relevant (cued) location was present in the neuronal population.

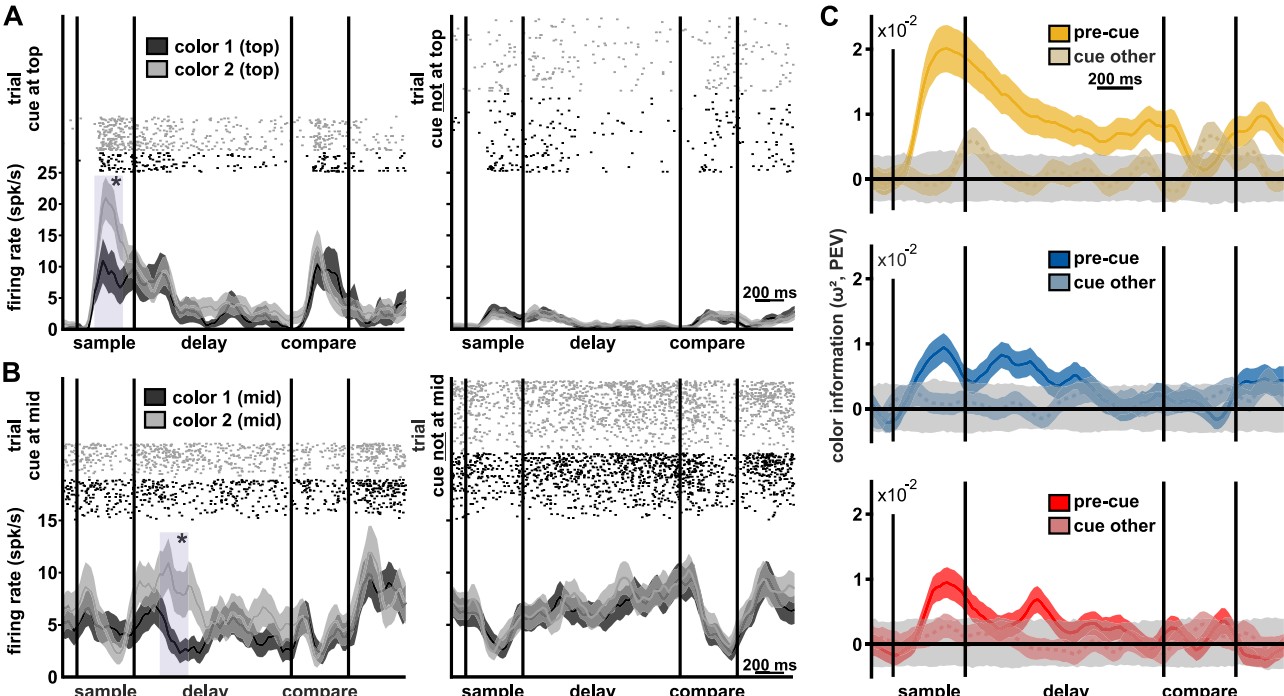

**Fig. 3 | Neuronal activity related to color. A** Dot-raster plots and spike-density functions of an example neuron. Left: The example neuron shows difference in spike density between color 1 and color 2 (at top location) when cue is at top location (trials $n_{color1} = 21$, $n_{color2} = 37$). Right: Same neuron, activity for colors 1 and 2 at top location, when cue was not displayed at top location (trials $n_{color1} = 63$, $n_{color2} = 67$). Purple shading indicates a significant difference in firing rate. Upper part: dot-raster plot of neuronal spiking, each dot represents an individual spike, sorted and color coded based on color identity at the top location in the session. **B** Same as in (**A**) for a neuron showing a difference during the delay at the middle location (trials with cue $n_{color1} = 52$, $n_{color2} = 44$; trials when cue was not displayed at middle, $n_{color1} = 100$, $n_{color2} = 96$). **C** Color information in neuronal population (quantified as PEV,

$n = 356$). The population of neurons encoded information about color at top, middle, and bottom locations (from top to bottom) when cue was at the respective location (solid lines). In contrast, when cue was not at the respective location (i.e., cue was instead at a different location, "cue-other"), the neuronal activity had no information about color (dashed lines). Lines indicate the mean PEV, and shaded areas around the mean are the standard error of the mean. The gray shaded area is the 95% interval [2.5 97.5] of PEV values attained after randomly shuffling the labels of color. Time periods when there is no overlap between a colored area and the gray area can be considered significantly different at $p < 0.025$ (based on permutation testing).

## Neuronal population encodes the color information only at cued locations

Birds had to encode and maintain the color in WM in order to detect if it had changed at any of the locations. If the cue had the effect of directing attention towards the cued location, we expected there to be an increase in information about the color at that location. The encoding of color in the neuronal population appeared weaker than that of cue location (27% vs. 54% significant neurons for color and location, respectively, with overlap between the groups, Fig. 2B). Nonetheless, a substantial number of neurons still encoded color identity in combination with cue location. When a cue was shown at a location, neurons strongly differentiated between colors (Fig. 3A, B, left panels). For the example neuron in Fig. 3A: during the sample after a cue had appeared at the top location, firing rate significantly differentiated colors at the top location (peaking 200 ms after sample onset, $F(157) = 14.38$, $p = 0.0004$, $\omega^2 = 0.1874$). Similarly, for the example neuron in Fig. 3B: firing rate differentiated colors (at the middle position) in the early delay, when a cue had appeared at the middle position (peaking 240 ms after sample offset, $F(194) = 16.34$, $p = 0.0001$, $\omega^2 = 0.1378$). In contrast, the neurons showed no difference between colors when the cue was shown at another location (Fig. 3A. B, right panels, same bins as above, $F(1162) = 0.4770$, $p = 0.4908$, $\omega^2 = -0.0032$, and $F(1174) = 0.0346$, $p = 0.8526$, $\omega^2 = -0.0055$, respectively). At the population level, when the cue was at the location, information about color (quantified as PEV, same as for location) peaked during the sample phase (when colors were on screen) and then gradually reduced throughout the delay phase (Fig. 3C, bright colors). Information about color was virtually absent for a given location when the cue was not at that location (Fig. 3C, darker colors). This indicated that the presence of

the cue increased information about color at its location. To expand on this finding, we analyzed the trials in which no cue was shown.

## Cue increases color information during sample and delay

In no-cue trials, the animals had no prior information on which color may change (Fig. 4A). The animals' behavioral performance was affected by the omission of the pre-cue in these trials (see subsequent results below). Neuronal activity, however, followed the same patterns observed in pre-cue trials. The firing rates of individual neurons (e.g., Fig. 4B) differed between the two colors at a location when there was no cue. For example, in no-cue trials, the neuron shown in Fig. 4B (upper axis) had peak differentiation at 440 ms after sample offset ($F(1147) = 16.97$, $p < 0.0001$, $\omega^2 = 0.0968$). However, color differentiation was higher when there was a pre-cue at a given location, as seen for the same time-point in pre-cue trials (Fig. 4B lower axis, $F(1111) = 26.91$, $p < 0.0001$, $\omega^2 = 0.1865$). Thus, the cue drastically increased information about the color encoded by neuronal firing ($\Delta = +0.0897$), compared to when there was no cue at that location. We confirmed this attentional shift effect of the cue at the neuronal population level by quantifying the overall attentional information shift ($\Delta$PEV). Both birds showed an increase in average information about color during the sample and early delay phases in pre-cue trials (Fig. 4C, Wilcoxon signed rank test and Bayesian one sample Wilcoxon signed rank test, see Table S1 for statistical results and Statistics and Reproducibility for details). The effect size of the increase at the population level was small (see Table S1). The overall attentional shift originated from information-gain ($\Delta$ information between no-cue and pre-cue, Fig. 4D and Table S2), and information-loss ($\Delta$ information between pre-cue at the analyzed location compared to pre-cue at a different location, Fig. 4E and Table S3), to similar degrees.

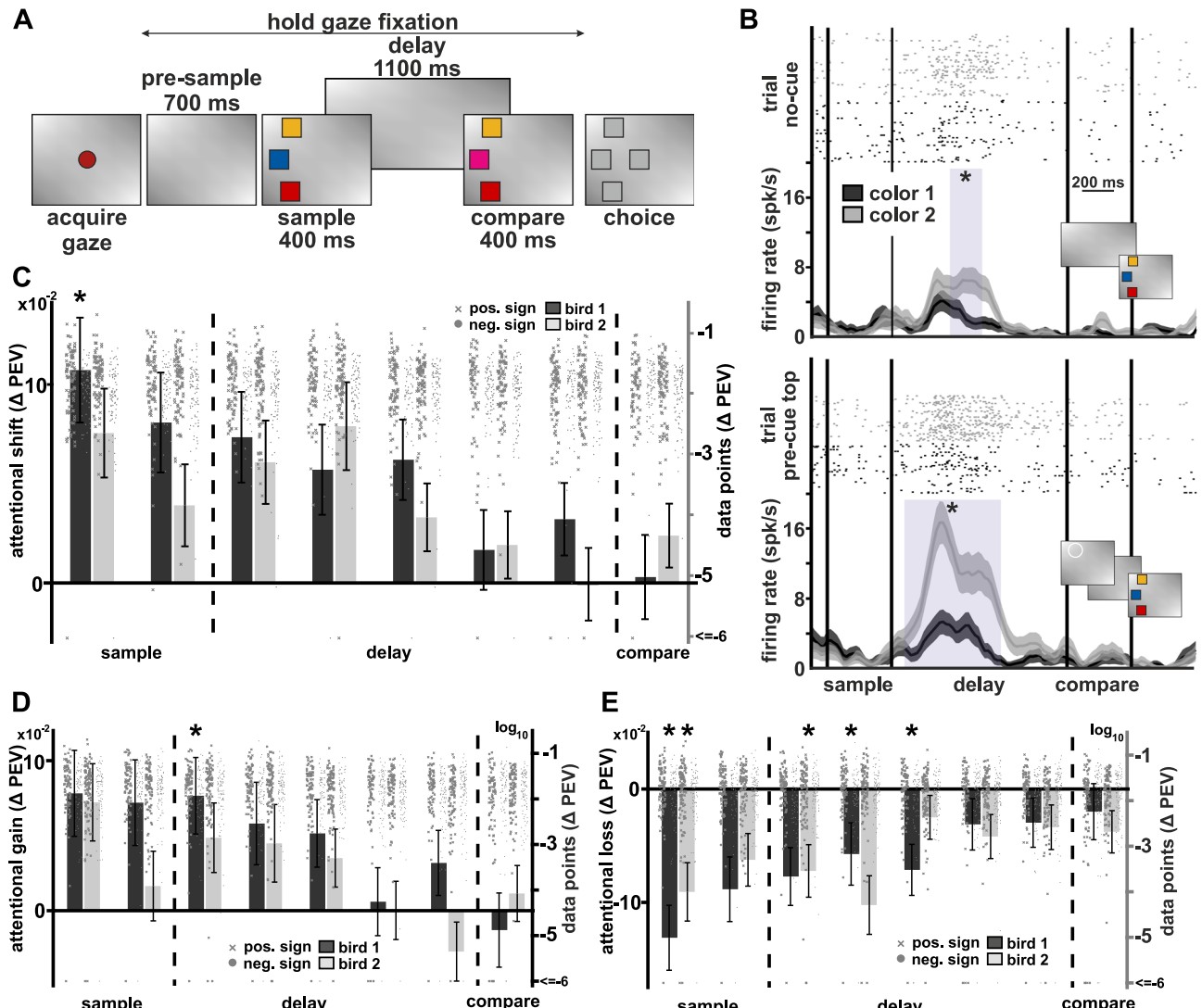

**Fig. 4 | Effect of attention on color information. A** Behavioral protocol of no-cue trials, same as in Fig. 1A, the only difference being the lack of a cue phase. Due to a lack of a cue the birds had to maintain color information about all three locations. In case none of the colors changed between the sample and comparison phases, the animal had to choose the center location. **B** Dot-raster plots and spike-density functions of an example neuron. The example neuron shows differentiation between colors at the top location. The presence of the cue increased the differentiation between colors. Top: neuronal activity in trials without cue ($n_{color1} = 82$, $n_{color2} = 67$). Bottom: neuronal activity of the same neuron in trials when the pre-cue had appeared at the top location ($n_{color1} = 59$, $n_{color2} = 54$). Purple shading indicates a significant difference in firing rate. **C** Effect of attention on color information. When birds attended a cued location, they had more information about color at that location. The attentional shift was quantified as the absolute average of cue dependent gain of color information ($\Delta$ PEV, [pre-cue minus no-cue]) and loss of color information ($\Delta$ PEV, [pre-cue minus cue-other], see also Fig. 3C). Binned in 200 ms bins, advanced in 200 ms steps along the time course of the trial. **D** Gain of information about color when cue was at the analyzed location, relative to when there was no cue. **E** Loss of information about color when cue was not at the analyzed location. Error bars in (**C**, **D**, **E**) indicate the standard error of the mean. Asterisks indicate significant difference from 0, based on a sign-rank test using Bonferroni correction. Each plot shows decadic log transformed datapoints ($n_{bird\ 1} = 163$ units, and $n_{bird\ 2} = 193$ units, per bin). Positive sign datapoints indicated with "x", negative sign data points indicated with "o"; refer to right hand log axis for values and Statistics and Reproducibility for more details.

## Behavioral signatures of selective attention in no-cue trials

In no-cue trials, when birds had to memorize all three colors (Fig. 4A), behavior and performance were strongly affected by the location of a color change. Both birds still had near perfect correct rejection rates (Fig. S2A, E, bird 1: Mdn 98.08%, 25th and 75th percentiles at 91.44% and 100.00%, respectively; bird 2: Mdn 100.00%, 25th and 75th percentiles at 93.59% and 100.00%, respectively). However, bird 1 appeared to produce at best around chance-level hit rates for each of the change locations with a substantial session-by-session variation (Fig. S2B, F, Mdn 37.5%, 22.73%, and 40.74% for top, middle, and bottom location, respectively). In contrast, bird 2 showed near perfect hit-rates for changes at the top location but missed virtually all changes at the middle and bottom locations (Fig. S2C, G, Mdn 100%, 3.65%, and 3.10% for top, middle, and bottom location, respectively).

To quantify if any of the three locations were preferred, we subdivided each session into five blocks of correct trials and calculated the hit rate in each of the blocks. We defined "preferred location of the block" to be the one for which the hit rate was highest. We then counted how many times (out of the five) a location was the preferred location. The location with the highest count was then considered to be the (surrogate) preferred location of the entire session (refer to methods section on behavioral preference analysis for further details). In this way, a medium and a least preferred location were also determined (based on the lower total counts, ties being ordered by location on screen, i.e., middle before bottom). Applying this new sorting on the performance data revealed similarities between bird 1 and bird 2. For bird 1 hit rate for the preferred location (Fig. 5A, Mdn 66.67%, 25th and 75th percentiles at 53.00% and 79.46%, respectively) was higher than for the least

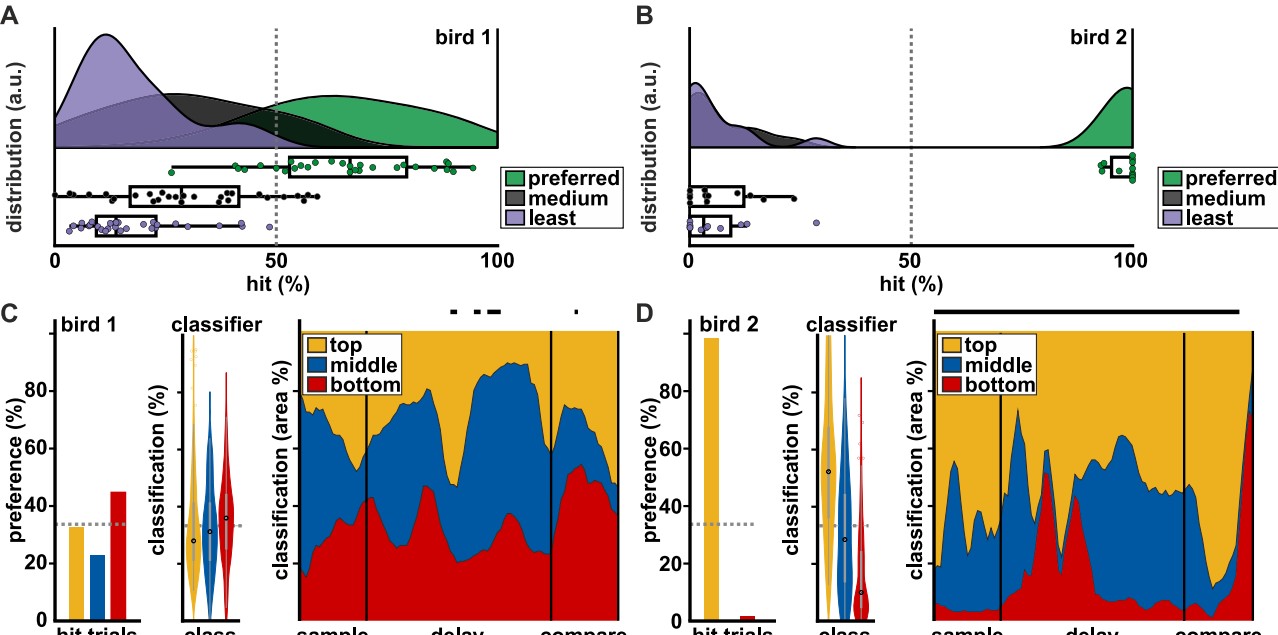

**Fig. 5 | No-cue trial performance and classification. A** Performance of bird 1 in trials with a color change. Plots as in Fig. 1B. Preferred location, medium location, and least preferred location of bird 1, following hit-rate classification. Lower plots show all data points ($n_{bird1}$ = 28 sessions); boxes indicate first quartile, median, and third quartile. Whiskers extend to 1.5* the interquartile range. **B** Same as in (**A**) for bird 2 ($n_{bird2}$ = 12 sessions). **C** Bird 1 had no consistent preferred single location. Bars: Behavioral preference for locations in no-cue trials, based on preferred location in behavioral block (single value per bar, visualization as bar plot for ease of comparison to classifier, see Fig. S1D, H for second and third location). Violin plots: Percentage of classification of no-cue trials, based on pre-cue training trials. On average the neuronal activity of bird 1 was equally classified as either top-, middle- or bottom-cued, consistent with the lack of a clear location preference. We used the

same SVM with the same training trials to classify the neuronal activity of the population ($n_{bird1}$ = 75 units) in no-cue trials (average across the entire trial period from pre-cue until choice on; boxes indicate first quartile, median, and third quartile. Whiskers extend to 1.5 * the interquartile range, and circles outside of whiskers indicate outliers). Area plot: No-cue trial classification across time between the sample and compare phase (black lines at the top indicate significant differences in classified location, based on a $X^2$-test). **D** Same as in (**C**) for bird 2 ($n_{bird2}$ = 110 units). The animal had almost exclusively the top location as its preferred location. In line with its behavioral preference, the classification of the unlabeled no-cue trials was strongly biased towards the top location (yellow) over the other two locations (blue and red), compare classifier values against expected values (dashed line).

preferred location (Mdn 13.79%, 25th and 75th percentiles at 9.34% and 22.81%, respectively) with the medium location in between (Mdn 28.57%, 25th and 75th percentiles at 16.97% and 41.54%, respectively). This difference was highly significant ($X^2(284)$ = 51.31, $p < 0.0001$), and post hoc comparisons confirmed that the preferred location had higher performance than both medium- and least-preferred (both $p < 0.0001$), whereas there was no difference between medium- and least-preferred ($p = 0.0958$). For bird 2 the reordering of locations had no effect on the performance values per location, as it consistently had the top location as its preferred location (Fig. 5B, $X^2(233)$ = 24.05, $p < 0.0001$, post hoc comparisons to the medium and least preferred location, both $p < 0.0001$, and no difference between medium and least preferred, $p = 0.9592$). Thus, the median hit rates did not change when compared to the location sorting (compare Figs. 1C and 5B).

**Neuronal activity in pre-cue trials predicted location preference in no-cue trials**
We found evidence of a preferred location for both birds, even though our surrogate approach to find the preferred location lacks sensitivity towards detecting a trial-by-trial change in preference. The presence of this strategy offered the possibility to analyze our neuronal dataset from a novel perspective: is the neuronal activity representing top-down attention different when an animal attends a location based on an external cue and when it attends it purely following an internal preference? We predicted that neural activity would be similar between pre-cue trials and no-cue trials if the cued location matched the location endogenously selected by the birds. Therefore, we should be able to use the SVM trained on the pre-cue trials to decode the endogenously selected location in no-cue trials. We could then compare this output of the SVM to the location of endogenous attention that we had

inferred from the animals' behavior (refer to methods section on behavioral preference analysis for details). Furthermore, the difference in location preference between the animals should be reflected in decoding performance. Bird 1 had no consistent preferred location within and between sessions (32.41% top, 22.76% middle, and 44.83% bottom, Fig. 5C, bar plot). In contrast, bird 2 consistently preferred the top location in no-cue trials (98.33% top, 0% middle, and 1.67% bottom, Fig. 5D, bar plot). We hypothesized that if we tested our classifier on no-cue trials, it should show a high percentage of top location classification only for bird 2, but no clear classification for any of the three locations for bird 1. We tested this hypothesis by using the pre-cue trials as a training-set (for which our earlier classification yielded performance significantly above chance, Fig. 2E, F), and the no-cue trials (of the same time bins) as a testing set. As for the pre-cue location classification, only neurons with significant information about location were used. No-cue trials of bird 1 were on average classified equally into one of the three locations (Fig. 5C, violin plot); unequal classification only happened during short phases of the delay (Fig. 5C, area plot, all $X^2(2, N=40) \geq 8.45$, $p < 0.05$ and comparison phase, all $X^2(2, N=40) \geq 7.55$, $p < 0.05$; equal classification during the rest, all $p > 0.05$, all $p$ values corrected for multiple comparisons, see Table S4 and S5 for all exact statistical values). In contrast, trials of bird 2 were predominantly classified as if there had been a pre-cue at the top location (on average at a rate of 53.75%, Fig. 5D, violin plot). This happened during the sample phase, the delay, and until the late comparison phase (all $X^2(2, N=40) \geq 6.35$, $p < 0.05$, Fig. 5D, area plot). Thus, training the classifier with neuronal activity of pre-cue trials was sufficient to decode preferred locations in no-cue trials in bird 2, when no external cuing was available. Furthermore, the lack of a clear location preference of bird 1 was reflected in a lack of clear cross-trial classification.

Given the behavioral differences between the two birds, we expected to find differences in how the cue influenced information for either bird. For bird 1, we expected to find a benefit for all locations, given its lack of a consistent preferred location. In contrast, if bird 2 indeed attended to the top-most location by default in no-cue trials, then cueing that location should not further increase information. However, cuing either the middle or bottom location should lead to increased information. We were able to confirm our expectation: For the most-preferred location, bird 1 showed a small increase in color information in cue trials compared to the same location in no-cue trials, while bird 2 did not gain additional information about color (Fig. S2I and J). For the least preferred location, both birds had moderate gains in information (Fig. S2K and L). Thus, consistent with the observed behavior and the employed behavioral strategy, the most-preferred location was barely affected by the external cue, whereas the least-preferred location benefited from the cue.

## Discussion

In line with previous results[49] crows demonstrated near perfect performance when attention was externally guided by the pre-cue and generally optimized cognitive effort by attending single locations. Similar procedures have reported the effectiveness of such attentional cuing using Posner's spatial attention task[50] in crows[18,19], and across visuo-auditory sensory modalities in barn owls[20]. Our two birds further showed signs of endogenously driven attentional selection in the absence of a pre-cue. We observed differences in behavior between the birds that were also reflected in the time course of their respective neuronal responses (Fig. 2D, E). When analyzing hit-rates in no-cue trials we found evidence that both birds attended only one location on a given trial. While bird 1 changed this attended location several times in each session, bird 2 consistently selected the top location. In the present protocol, the animals were presented with three colors in one visual hemifield, placing high demands on WM. Crows have a capacity limit of about two items per hemifield; exceeding this limit results in reduced accuracy[51]. Focusing attention on only one location allowed the animals to keep WM load to a minimum while still receiving rewards in two-thirds of trials. To be more specific, maintaining only one color at an actively attended location, instead of maintaining three colors at three locations, allowed reporting a change at the attended location with relative ease (virtually guaranteeing a correct response in one out of six conditions). At the same time, it guaranteed to successfully report all trials without any change, at any location (virtually guaranteeing a correct response in an additional three out of six conditions). Consequently, incorrect responses were limited to trials where a change occurred at one of the two unattended locations (i.e., the remaining two out of six conditions). In terms of signal detection, the birds optimized correct rejections at the cost of a reduced hit rate in trials without a cue. This behavior was different from a previous study that varied the load on WM (up to six colors across both hemifields[49], and a change on every trial[51]). In that study, attending only one location would have resulted in much worse odds (depending on a number of colors on a given trial). Only when the animals were transitioned to the new protocol did focusing on one location become a viable strategy that the birds promptly utilized. Consequently, crows can flexibly adjust behavioral strategies to optimize effort-reward ratios in response to varying experimental designs[52]. We cannot deduce if the crows' choice of behavioral strategies was strategic in a *sensu stricto*, "designed (…) [to] achieve a particular objective"[53], or was simply an extension of the behavior in pre-cue trials, in which paying attention only to one location was already optimal. However, because their behavior indicates the direction of attention to a single location, we were able to study endogenous attention in the absence of any external cues.

Could exogenous capture of attention towards the pre-cue location be a plausible alternative explanation? Exogenous attention is commonly defined as involuntary, caused by an unreliable and uninformative stimulus[4,54], of short duration, less than 500 ms[4,55], in crows, specifically up to 400 ms[19]. In contrast, we used a fully reliable pre-cue that predicted the onset of the only relevant target color. Furthermore, attention to the localized target color had to be sustained throughout the trial until a behavioral

decision could be made (2400 ms after the offset of the pre-cue). Finally, the underlying attentional mechanisms in both pre-cue and no-cue trials appeared as interchangeable at the behavioral level. Thus, in the absence of the pre-cue (in no-cue trials), birds endogenously chose to attend their preferred location. Our observations at the neuronal level corroborate that the attentional effect on color information in pre-cue trials and in no-cue trials was the same. Ideally, we would have been able to analyze a consistently high number of correct no-cue change trials across all three locations in all sessions. This would have allowed us to directly confirm the similarity of the neuronal attention signal between pre-cue trials and no-cue trials for any given location. Nonetheless, three aspects of particular importance support our interpretation that the neuronal activity reflects endogenous attention, based on the direct observations of neurons of bird 2. First, we did not find expected load differences in information between pre-cue trials (one item) and no-cue trials (three items[46,56]). Instead, color information was either high, for an attended location (either following the pre-cue or a preference in no-cue trials), or low for an unattended location (any location without a pre-cue in pre-cue trials or any non-preferred location). Considering the behavioral strategy, the two levels of color information make sense, as only one color was attended and thus its neuronal representation enhanced, while the others were likely inhibited, even in no-cue trials. Second, bird 2 exhibited minimal difference in color information for the preferred top location between no-cue trials and pre-cue trials, unlike any non-preferred location, indicating the same attentional processing in both trial types. Third, the overall neuronal activity in pre-cue and in no-cue trials was in fact so similar that our classifier consistently decoded the top location from no-cue trials, in line with the behavioral strategy. In contrast, there was no consistent decoding of the preferred location of bird 1, owing to the lack of a consistent location preference across sessions.

We cannot exclude the possibility that overt attention led to foveation and subsequently changed sensory (bottom-up) processing of color. However, we consider possible confounding effects of retina-based changes in processing to be limited. In contrast to primates, eye movements are only of small amplitudes in birds[57–59], but see in ref. 60 for an exception. In laterally eyed species, such as crows, larger amplitudes of eye movements happen mostly in the relatively small binocular frontal range, surrounding the area where the beak is[58,61]. In contrast, when directing their gaze to objects in more lateral parts of the visual field, birds make use of their flexible neck, with a fast head orientation towards the attended object[59]. Our camera tracking ensured a central position of the head and that there was only minimal head movement. Additionally, our tracking guaranteed that only one eye had visual access to the stimuli (i.e., presentation outside the area of binocular overlap[61,]). In primates, foveation significantly influences color processing, primarily because the required photoreceptor cones are much more abundant in the fovea than in more peripheral regions of the retina[62], and have a nearly one-to-one transmission to bipolar and ganglion cells[63]. This causal reasoning is only partially applicable to birds[23]. Corvids have a fovea with a heightened density of cones, suggesting enhanced color sensitivity[64–68]. However, differences in their ganglion cell density between the fovea and periphery stays within the same order of magnitude[59,66,69], while that of primates differs by three orders of magnitude[70]. Thus, bottom-up changes in color processing between foveal regions (overt attention) and more peripheral areas of the retina (covert passive viewing) should be less pronounced than in primates. Finally, using more centralized or symbolic cues instead of purely spatial cues, e.g., an arrow pointing toward the relevant location, commonly done in human studies[50], would not meaningfully change the underlying process. For crows, both a spatially coincident cue[19], and a centralized cue[18] have been used previously to disambiguate endogenous and exogenous attention. Thus, with adequate training, crows can learn how to use spatially non-overlapping cues, to direct endogenous attention and likely can learn how to use symbolic ones through associative learning. Neuronal recordings of visual regions will be required to further disambiguate the nature of the neuronal correlates in NCL (see below).

In addition to reciprocal connections with higher order sensory areas, the NCL also has descending projections to premotor regions in the arcopallium[71–73]. There, the arcopallial gaze field[33], comparable to primate frontal eye fields[33], is directly connected to the midbrain attention network, which is highly relevant for orientation movements and stimulus selection[8,32,34,74]. It is thus conceivable that activity in NCL could be related to motor activity of directed eye movements, gaze fixation, or pecking. While we cannot disambiguate these different processes in our recordings, we think it is unlikely that our observations could be explained more parsimoniously by purely motor-related activity. Differences in neuronal activity could be correlated to eye-movement towards or gaze fixation on different cued locations. However, color information (firing rate differences for different sample colors) at a cued location cannot simply arise from such motor-related signals. Furthermore, the birds were only able to make a planned motor movement towards the middle (no-change trials) or to the lateral keys (change trials) after a mandatory comparison phase. We analyzed the data for change-detection (and subsequent peck-location) and confirmed that such a signal was not present prior to the choice period.

Our results indicate that there is a neuronal correlate of attention in NCL, a higher associative region of birds comparable to the PFC of mammals[26]. The NCL lacks the distinct layering of the mammalian neocortex[75]. Thus, neocortical layering is not a pre-requisite for endogenous attention. In line with previous findings on avian WM, neurons in the NCL maintained information about colors[45,46,76] and different spatial locations of the pre-cue[38]. The amount of information per maintained item in WM (e.g., color) diminishes as the total load of items in WM increases. Information quality is highest at a load of one item and subsequently declines when more items are added to WM, following divisive normalization of population activity[46,56,77]. We found that information quality about color in pre-cue trials was virtually identical to no-cue trials. Despite the different load requirements (one item in pre-cue trials and three items in no-cue trials), we found neuronal activity resembling a WM load of only a single item[46]. Cross-trial transfer of classifier performance further corroborated the similarity of neuronal activity. Although crows can maintain up to three items simultaneously[51], here, they chose to selectively attend to only one location. Thus, we can quantify the difference between attended and unattended colors but not between an attended color and a neutral multi-color array (i.e., a WM load of three items without directed attention). When primates engage in similar paradigms, directing attention towards a visual stimulus enhances the neural representation of this stimulus, i.e., there is an increase of conveyed information[78]. Importantly, multiple cortical regions, including sensory cortices, are involved with different contributions[3,78]. Associative regions in the frontal and parietal cortex are important higher-order nodes of the attention network that exert top-down influence on sensory cortices[79,80]. We interpret that the neuronal correlates of pre-cue location and enhanced color information we observed in NCL were functionally similar to neuronal activity in primate PFC, representing a top-down signal[81,82]. However, more research on the dynamics of attention-related activity in the avian telencephalon is required to unambiguously describe the role of such signals[79,83]. Top-down signaling of NCL would influence sensory regions of the visual pathway, comparable to how frontal and parietal cortical regions interact with sensory cortices[3,84]. It will be necessary to investigate inter-regional activity in the bird telencephalon to better understand how closely non-layered regions like NCL resemble their presumed functional cortical equivalents in mammals[1,26] and to better qualify how top-down control of attention may shape sensory processing. Prime targets for recordings would be the nidopallium frontolaterale (NFL) and the entopallium (ENT). Each of these visual regions has direct reciprocal connections to NCL[71,73], and each other[73], and shares neuronal circuit architecture with the mammalian cortex[75]. Furthermore, both NFL and ENT have been shown to process color information (and complex visual stimuli) in pigeons[76,85–87]. Neurons of NFL are involved in learning of visual stimulus-outcome associations[76,85,88], which likely are targets of attention related processes to optimize behavior. The ENT may be of particular interest as it is a hub for visual processing that may be functionally comparable to parts of the extrastriate cortex of mammals[85], playing a crucial role in pattern discrimination in pigeons[89,90] and zebra finches[91,92]. ENT receives ascending projections from the midbrain stimulus selection network, relayed by the thalamic nucleus rotundus[93]. These projections may transmit multimodal signals to the ENT[94] that are involved in attentional control[95]. Thus, it is possible that top-down signals of NCL and ascending signals of the stimulus selection network converge in the ENT. Alternatively, and non-exclusively, descending projections of NCL to the optic tectum, relayed via the arcopallial gaze field[34,74], may play an important role in top-down control. Investigating these connections in detail may help to further understand the role of the avian tectum in (endogenous) attentional control.

In conclusion, crows demonstrated remarkable adaptability by efficiently focusing attention on single locations, a strategy effective in both externally guided and autonomous, endogenous contexts. This finding underscores their ability to optimize effort-to-reward ratios, showcasing cognitive flexibility previously underappreciated in avian species. The neuronal correlates of endogenous attention we found in neurons of NCL further highlight the role of attention in executive function and the similarity of higher associative regions in mammals and songbirds.

## Materials and methods
### Animals
For this study, we worked with two hand-raised adult (3 years of age) male carrion crows (Corvus corone corone). The birds were housed in social groups in spacious aviaries. During the study, the animals had *ad libitum* access to water and were held on a controlled food protocol. Both birds had previously participated in a behavioral study investigating control over WM[49]. All experimental procedures and housing conditions were carried out in accordance with the National Institutes of Health Guide for Care and Use of Laboratory Animals and received ethical approval by the national authority (LANUV, protocol no. 84-02.04.2017.A001). We have complied with all relevant ethical regulations for animal use.

### Experimental setup—hardware
We trained and recorded the birds in automated operant training chambers (50 cm wide, 50.5 cm deep, and 77.5 cm high). Each chamber was equipped with a 22-inch acoustic pulse touchscreen (ELO 2200 L, APR, Elo Touch Solutions Inc., Milpitas, CA), and an automated pellet feeder, which delivered a food reward following a correct trial. During data acquisition and training, the birds sat on a wooden perch in the middle of the chamber, facing the touchscreen at a distance of 7 cm (measured as screen to eye). The software used to automatically run the experiment and acquire data was executed by control computers connected to the chamber. Digital input and output of the control computers were handled by a microcontroller (ODROID C1, Hardkernel co. Ltd, Anyang, South Korea) running custom software connected through a gigabit network. We used an infrared camera (Sygonix, Nurnberg, Germany) mounted on the ceiling in the chamber for remote monitoring of the birds during experimental sessions and two additional cameras for head tracking (see below).

### Experimental setup—software
The behavioral protocol, camera tracking, and subsequent analysis were performed with custom code written for Matlab (The Mathworks Inc., ver. 2018b) in combination with available toolboxes, including the Psychophysics toolbox[96], and the Biopsychology toolbox[97].

### Head tracking
We tracked the birds' head position using two computer-vision cameras (Chameleon 3, Point Gray Research Inc., Canada), one positioned directly above the animal and the other positioned to their left. Each camera ran at a framerate of 50 Hz and tracked the position of a 3D-printed reflector that was attached to a lightweight custom head-post (<350 mg) on the bird's head during experimental sessions. Incoming positional data was smoothed by integrating over two frames, using custom code in Matlab running on a control computer in parallel to the behavioral paradigm and

electrophysiological recording. The camera tracking was used to ensure that the birds positioned their head in the desired location (acquire-gaze) and maintained that position throughout the trial (hold-gaze). Birds were trained to hold-gaze with no more than 2 cm horizontal or vertical displacement, at a stable angular rotation of no more than 17° in the horizontal or vertical plane.

## Stimulus design

The current study's stimulus presentation followed the design originally described in refs. 46,51. Colored squares (15 distinct colors) were displayed on the left side of the touchscreen with 10° of visual angle (DVA). The stimuli were positioned in three distinct locations, one on the horizontal meridian (middle-location), and each one at 45.8 DVA above (top-location) and below (bottom-location) the meridian. Relative to the center of the screen, the center of the square stimuli at the top- and bottom-location were at 55.4 DVA, and the middle-location was at 54 DVA. We used a white ring (cue) of 11 DVA to indicate the relevant position in a cue trial. The cue was positioned at the exact location of the color square that was tested in the given trial. We used head tracking to train the birds to stay within pre-defined ranges of horizontal and vertical position, as well as angular rotation of the head, to ensure that the birds were able to see the stimuli exclusively with their left eye. Carrion crows have a maximal binocular overlap of 37.6 DVA[61]. Therefore, the three locations we used for the color squares were all well outside the binocular range, as long as the bird stayed within the pre-defined ranges of the head-tracking. For each session, one pair of colors was assigned to each of the three locations. Each location had its own exclusive pair for that session. The pairs were randomly chosen from a pool of 15 colors. Figure 1A gives an example: the color-change occurred at the middle position where blue (B) is presented during the sample and magenta (M) during the choice. In this particular session, the middle location could thus show either of the following colors during the sample and choice: B-M (shown in Fig. 1A), M-B, B-B, M-M. In the subsequent session, a new random pair of colors was displayed at this location. The order of presentation of colors within a pair and the change location were randomized and balanced across trials so that each condition had an equal likelihood to appear.

## Behavioral protocol

Our behavioral protocol was an adjusted version of the protocol previously described by ref. 49. The birds performed a change detection paradigm with active head tracking. Each session consisted of 850 trials (with an approximated total duration of 180–200 min, including short breaks with ad libitum access to water). Following an inter-trial-interval (ITI) of 2000 ms, the birds were presented with a red disk at the center of the screen (initiation phase, INI) for a maximum of 20 s. Within the available time, the birds were able to initiate a trial by acquiring head fixation, i.e., moving the head to within the pre-defined range and remaining there for 40 ms. They were required to maintain their head in a stationary position (with a displacement of less than ±2 cm horizontally or vertically) and gaze directly at the center of the screen (with a rotation of less than ±17° horizontally or vertically) to initiate and successfully complete the trial[49]. When the birds successfully initiated a trial, the red disk disappeared. There were two different trial types. After initiation, a trial continued with a delay of 700 ms (no-cue, 50% of trials, Fig. 1A), or with the presentation of a pre-cue for 200 ms (pre-cue phase), followed by a 500 ms delay (pre-cue, 50% of trials, Fig. 4A). In pre-cue trials, the location of the cue was either at the top, middle, or bottom location. Cue location was pseudorandomized across trials (1/3 chance of cue appearing at any location, but no more than two consecutive trials with the same location). In both trial types, the sample phase followed. During the sample phase, three colored squares were shown for 400 ms. One color at the top, one at the middle, and one at the bottom location. The birds' task was to observe and remember the colors of all of the squares (in no-cue trials) or the color of the square in the previously cued location (pre-cue trials). The squares vanished, and a blank screen followed for 1100 ms (delay phase). After the delay, all three colored squares reappeared for 400 ms (compare

phase). In no-cue trials, there was a 50% chance that one of the three colors had changed. In cue trials, there was a 50% chance that the color at the cued location had changed. The end of the compare phase was indicated by all squares on the screen turning uniformly gray and a fourth square appearing in the center of the screen. If a color had changed, the birds were trained to peck any of the original three squares to indicate the change. Throughout the recording sessions, we observed that the birds virtually always pecked at the location where they had detected a change. If no color had changed, they were trained to peck at the square in the center. A correct answer, either detecting a change (hit) or detecting the lack of change (correct rejection), was rewarded with a food pellet delivered through the automated feeder. Any incorrect answer (miss and false alarm) resulted in a screen flash and a timeout (18 s). Failure to acquire head-fixation within the given time to initiate a trial resulted in the disappearance of the red disk and the start of the subsequent ITI. The birds had to hold the head position they had acquired during INI throughout the trial until they reached the choice phase. Failure to maintain the head position at any time point before the stimuli turned gray resulted in the abortion of the trial (gaze break), followed by a brief white screen flash and a time out (10 s). Thus, in all completed trials, the birds had a stable head position and were directing their gaze toward the center of the screen. Non-cognitive strategies, such as positioning the head at a specific location or rotating the head to see the stimuli with more than the left eye, were therefore not available.

## Animal training

Prior to participating in this study both birds had participated in very similar behavioral tasks using the same touchscreen and camera tracking setup. The basic details can be found in refs. 49,51. Both animals were transferred to the behavioral task after completing data acquisition in ref. 49, and were thus familiar with the touchscreen and head tracking and holding their head still in a specific position. The behavioral task was simplified by removing both the retro-cue and the three stimuli from the display. Both animals quickly adjusted their behavior to perform in the simplified protocol and were then implanted with microelectrodes for neuronal recordings.

## Surgery

The surgery protocol was identical to the one reported by ref. 46 and is briefly repeated here. Both animals were chronically implanted with a lightweight head-post to attach a small reflector during the experiments. Before surgery, animals were deeply anesthetized with ketamine (50 mg/kg) and xylazine (5 mg/kg). Once deeply anesthetized, animals were placed in a stereotaxic frame. After attaching the small head-post with dental acrylic, a microdrive carrying a microelectrode was stereotactically implanted at the craniotomy (Neuronexus silicon probe, 32 channels, Technologies Inc., Ann Arbor MI, DDrive). The silicon probe was positioned in NCL (AP 5.0, ML 13.0) of the right hemisphere based on histological studies on the localization of NCL in crows[45,98]. After the surgery, the crows received analgesics. Following the administration of analgesia (morphasol, 3 ml/kg), the animal was placed in a recovery cage until fully recovered.

## Recordings

We recorded single-unit activity from crow NCL during the execution of the change detection task. Signals were recorded in the right hemisphere using chronically implanted 32-channel microelectrodes with an inter-electrode distance of 50 μm (Neuronexus Technologies Inc., Ann Arbor MI). Signals were recorded at a sampling rate of 30 kHz, amplified, filtered with a bandpass (0.01 Hz–7.5 kHz), and subsequently digitized by an Intan RHD2000 head stage and a USB-Interface board (Intan Technologies LLC, Los Angeles, CA). Digital event codes were sent by the behavioral computer using a custom IO device (details available at https://www.ngl.psy.ruhr-uni-bochum.de/ngl/shareware/index.html.en) and were recorded in parallel with the electrophysiological data. Prior to each recording session, using the microdrive, all electrodes were advanced by 750 μm (half of the available active zone). Recording of the neuronal data commenced 20 min later. Signals were observed online during recording to check for the presence of

physiological data but were not further pre-selected for task involvement. Spike-sorting was performed offline using the semi-automatic Klusta-suite software[99] to efficiently separate individual neuronal clusters of spikes. For spike sorting, data was further filtered with a bandpass (0.5 kHz–7.125 kHz).

## Behavioral analysis of preference

We estimated the location preference of an animal from behavioral hit performance by dividing each session into five equal blocks of trials (i.e., non-overlapping fifths of completed trials, from beginning to end). Within each block, we calculated the hit rate of detecting a color change in no-cue trials. We then sorted hit rates from highest to lowest in each block, labeling them as "preferred," "intermediate," and "least preferred". To analyze the overall preferred location in a session (Fig. 5A, B, analogous to analyses per pre-cue location, shown in Fig. 1B, C) we counted how many blocks a given location was defined as "preferred". The location with the highest count was then set to be the overall preferred location. Ties of count were resolved by selecting based on the highest display location, i.e., giving precedence to the top and middle locations over the bottom location. For the more detailed analysis comparing preferences to the classification of neuronal activity (Fig. 5C, D), we calculated the overall preference as the fraction of all blocks of all sessions, out of all blocks of all sessions. For example, bird 2 had 12 sessions, each divided into five blocks, for a total of 60 blocks. Preference for top, middle, and bottom locations was then counted and divided by 60 to yield a block-wise preference for each location. The determined preferences were stable over a wide range of selected block sizes ($s = 3$ to $s = 10$, where block size was 1/s of trials in each session).

## Statistics and reproducibility

We analyzed data from 28 sessions of bird 1 and 12 sessions of bird 2. The disparity in session numbers was a consequence of the smaller session-by-session yield of recorded single unit activity in bird 1. We analyzed only complete trials, i.e., those in which the bird made either a correct or incorrect response. Trials in which the bird failed to maintain voluntary head-fixation at any time point were immediately aborted and were not analyzed. Consequently, the number of analyzed trials differs between birds and sessions. We analyzed an average number of 456 trials for bird 1 (range between 343 and 573) and 402 for bird 2 (range between 322 and 536).

The inclusion criterion for electrophysiological analyses was a minimum number of 20 trials per tested condition and a minimum of 0.5 spikes per second average firing rate of a single unit during the ITI. We analyzed a total of 356 recorded units without further post-hoc exclusions.

We used either common parametric (*t*-test, ANOVA) or non-parametric (Kruskal–Wallis, $X^2$-test) statistical tests, in accordance with data being normally or non-normally distributed, respectively. Instantaneous firing rates of neurons were computed in 200 ms bins, in a sliding window across the analysis interval at a step size of 20 ms; whenever consecutive bins were used in a test, we ensured that there was no overlap between bins. For all analyzes that return *p* values, a significance level alpha = 0.05 was set. Statistical judgments about significance were obtained after correcting for multiple comparisons so that the corrected alpha was retained at 0.05 (either using the Bonferroni method or by adjusting the alpha level directly). For the analysis of the neuronal activity, to avoid biasing results towards any particular effect, we performed populational analyzes on all recorded neurons that met the inclusion criterion. Due to unbalanced numbers of trials between conditions, we subsampled each condition and analyzed only the first 20 trials of any condition. This ensured comparable results between analyzes unaffected by sample size. We further focused on analysis of the effect size of an analysis factor, measured as the percent explained variance (PEV), either as $\omega^2$ (Eq. 1) or as $\omega_p^2$ (Eq. 2) where applicable.

$$\omega^2 = \frac{SSQ_e - (df_e * MS_{err})}{SSQ_{total} + MS_{err}} \tag{1}$$

$$\omega_p^2 = \frac{df_e * (F_e - 1)}{df_e * (F_e - 1) + n} \tag{2}$$

Both equations use the output of an ANOVA, where $SSQ_e$ is the sum of squares of factor e or total, $MS_{err}$ is the mean squares of error term, $F_e$ is the F-statistic of factor e, $df_e$ are the degrees of freedom of factor *e*, and *n* is the number of observations.

The PEV can be interpreted as the amount of information encoded by a given factor. We employed permutation testing to judge if the amount of information exceeds what would be expected were the data random. To do so, we randomly shuffled the data labels (i.e., the grouping of the data) before performing the calculation of the effect sizes. This process was repeated 1000 times to obtain a distribution of PEV values that can act as a null distribution against which the values obtained from the real dataset can be compared. We considered the results to be significant if their value exceeded 95% of values generated by permutation.

For visualization purposes, when individual data points of PEV values per neuron are shown alongside summary statistics in bar plots (Fig. 4), the raw datapoints were transformed by first grouping them based on their sign (positive and negative), the negative sign was then removed and values transformed to a decadic logarithmic scale with a separate axis (e.g., a value of $-0.001$ was transformed to $10^{-3}$ and plotted with the indicator for a negative sign). Values smaller than $|10^{-6}|$ were set to $10^{-6}$ for visualization purposes. All statistics were performed on the original values.

Bayesian analysis for the attentional shift at the population level (Fig. 4C) was conducted using JASP (ver. 0.19.1)[100], following suggestions made by ref. 101. We used the Bayesian one-sample T-test module, the Wilcoxon signed rank test with 1000 samples, and a default prior (Cauchy, scale 0.707). We tested the alternative hypothesis that population values were > 0 against the null hypothesis that they were not (i.e., one-sided). We report all details on the support for the alternative hypothesis using the Bayes factor ($BF_{+0}$) in the supplementary material, alongside the frequentist statistics (Table S1).

## Classifier analysis

We investigated information about cue location encoded and maintained by the neuronal population using multi-class support vector machine (SVM) classifier models[102,103]. The SVM was trained and tested on firing rates of all recorded neurons in an analysis interval spanning from the onset of the cue until the onset of the choice phase (when the stimuli turned gray, Fig. 1A). Instantaneous firing rates of neurons were computed in 200 ms bins, in a sliding window across the analysis interval at a step size of 20 ms. For this analysis, firing rates were then z-scored based on the populational firing rate and standard deviation in each bin.

Our approach followed the classifier analysis outlined in ref. 42. The available set of trials for analysis was constructed as a pseudo-simultaneous population of neurons, encompassing firing rates for 120 pseudo-simultaneous trials. For classifier training, a random subset of 90% of the available trials was used (i.e., 108 trials, 36 trials per condition). Subsequent testing of the trained classifier was performed with the remaining 10% of trials (i.e., 12 trials, 4 trials per condition). This yielded a classification of the test trials into one of the three available conditions (i.e., cue location top, middle, or bottom), which was then compared to the actual labels of the testing trials. The comparison indicated the performance of the classifier. This process (random drawing of trials for training and testing, calculation of achieved performance) was repeated 1000 times, generating a distribution of performance per bin of which the mean and the standard error of the mean is reported. To estimate a baseline performance of classification (the range of attainable classification performances under random conditions), we performed the procedure described above 1000 times using shuffled data labels (i.e., where neuronal activity was not associated with a particular trial type). This produced a distribution of classification performance values per bin, centered around the theoretical chance level of one-third (classification into one of three groups). The training-testing procedure was implemented

to test bins of neuronal activity at any other time point throughout the interval of interest by training in one bin and testing all other available bins (cross-temporal analysis). Classification performance values along the diagonal of the resulting train-test matrix (i.e., where training bin was equal to testing bin) were considered significant if the attained performance values exceeded the range of values attained from the randomized labels run (i.e., when the inner 95% of values in each distribution did not overlap). This procedure yields a conservative estimate of significance, at least at $p < 0.025$. For cross-temporal training-testing, clusters of significant test bins (at alpha of 5%) were considered significant if clusters exceeded a size of five neighboring bins.

## Reporting summary
Further information on research design is available in the Nature Portfolio Reporting Summary linked to this article.

## Data availability
Source data that support the findings of this study are available at Zenodo (https://doi.org/10.5281/zenodo.13788459)[104].

## Code availability
Custom Matlab code for analysis has been deposited and is available at Zenodo (https://doi.org/10.5281/zenodo.13788459)[104].

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

## Acknowledgements
We would like to thank Jesus J. Ballesteros and Robert Schmidt for their helpful comments on an earlier version of the manuscript. This work was supported by Volkswagen Foundation Freigeist Fellowship 93299 (JR) and Deutsche Forschungsgemeinschaft (DFG, German Research Foundation) Project A19 of the collaborative research center SFB1280 Projektnummer 316803389 (JR).

## Author contributions
The authors L.A.H. and E.F. have contributed equally to the manuscript. L.A.H.: Conceptualization, Software, Investigation, Formal analysis, Data Curation, Visualization, Writing—Original Draft, Writing—Review & Editing; E.F.: Conceptualization, Data collection/acquisition, Data Curation, Investigation, Methodology, Software, Surgery, Writing—Review & Editing; J.R.; Conceptualization, Funding acquisition, Methodology, Project administration, Surgery, Resources, Writing—Review & Editing.

## Funding

## Competing interests
The authors declare no competing interests.
