## [Transparent Peer Review file · Communications Biology]

Neuronal correlates of endogenous selective attention in the endbrain of crows

Corresponding Author: Dr Lukas Hahn

Version 0:

Reviewer comments:

Reviewer #1

(Remarks to the Author)

Neuronal correlates of endogenous selective attention in the endbrain of crows

Summary

The authors investigate an important research question, the neuronal correlates of attention in a diverse animal model – the crow. Hahn and Fongaro et al., trained 2 crows on a visual change detection task where 3 differently-colored squares were presented top, middle, and bottom during the sample phase. After a delay in which the screen clears, the 3 squares reappear where one of the squares with 50% probability could have changed color and crows have to indicate whether or not this change occurred. Critically, this paradigm was applied to two different trial types. In the first, a pre-cue (white circle) indicated the position in which the square would change color if it did so and thus crows only had to attend to the colored square at one spatial location. In the second no-cue condition, crows should have attended to color of the squares at all three spatial locations. Crows were able to perform the task well in the pre-cue trials and recordings in the nidopallium caudolaterale (NCL) revealed enhancement of color information at the cued location. Interestingly, crows appear to leverage the structure of the task in the no-cue condition and instead of attending to all 3 squares with some difficulty given working memory constraints, picked one to attend to during the trial. Crow 1 changed preferences and crow 2 preferred the top square. Authors were able to capitalize on this behavior to examine differences in neural coding when squares were externally cued and when they were internally preferred by the crows. The project is well-motivated, the analyses are thorough, and I have a few comments I would like addressed.

Major Comments:

It is unclear to me in the no-cue trials, why crows have such a high performance in reporting no change. In the discussion (Lines 401-403), the authors write ‘memorizing only one instead of three colors allowed reporting a change at this location with very high performance (1/6 conditions) and to successfully report all trials without change (3/6). Why is it easier to detect if no change had occurred if crows were attending to only one stimulus leading to ‘near perfect correct rejection rates)?

Related to the above point, in the methods (Lines 646-648), ‘If a color had changed, the birds were trained to peck any of the original three squares to indicate the change. If no color had changed, they were trained to peck at the square in the center.’ If crows could peck at any of the three squares, did they tend to peck at the square in which the change occurred or at the square that they were presumably attending to?

In the section, Neuronal activity in pre-cue trials predicted location preference in no-cue trials, the lack of a clear location preference of bird 1 was reflected in a lack of clear cross-trial classification. However, this appeared to be from throughout a session. In a previous page (Line 301), authors calculated a ‘preferred location of the block’ for bird 1. If analyses were done on a block level for this bird, would there be change in decoding performance when the data is not organized by spatial location but by preference? This would further support the discussion that the same attentional processing is occurring in both trial types (Line 434-439) which is currently focused on bird 2.

The title and text reference ‘endogenous attention’ and the authors write that ‘Exogenous attention is commonly defined as involuntary, caused by an unreliable and uninformative stimulus, of short durations [...]. In contrast, we used a fully reliable pre-cue that was predicting the onset of the only relevant target color.’ However, the pre-cue was present at the same spatial location as the square to be attended to. Could the authors discuss what might occur if the pre-cue was centrally

located instead?

There is little text on the training procedures of how the crows learned this task from a previous study (Fongaro & Rose, 2020). Could you elaborate on different training stages to shape the final behavior? Are there differences in the amount of training between crows?

Minor Comments:

The figures are a bit cluttered with black outlines around each subfigure.

In Figure 1, the timeline of the task was initially a bit confusing with the different sizes of the grey rectangles arranged before and behind each other.

In Figure 2, 'Purple shading indicates periods when spike-density between different cue locations was significantly different'. It appears in C that there is purple shading even before the onset of the cue?

In Figure 2, can the authors speculate on why in Bird 1, there is no significant cluster for decoding cue location in the pre-cue period? And more generally, why the temporal consistency between phases seems limited (Line 208), which I would not expect given that 'endogenous attention [is] sustained (Line 45)'.

In Figure 2, why does there appear to be a division in the cue information present (e.g. classifier performance) between the beginning and the end of the delay phase in both birds (E and F)?

Do crows have a preference for the color they would like to attend to out of the 15 possible colors when examined together? Is that what is calculated for color in 2B or is that just between the pairs of colors (Figure 3)?

There were 850 trials in the session and sessions were broken into blocks of 5 for the preference calculation. How was this block size chosen? Are the preferences robust if the block size is adjusted to be smaller or larger?

Reviewer #2

(Remarks to the Author)

This paper investigates the neural correlates of endogenous attentional orienting in birds, using crows as a model species. Extracellular single unit recordings were performed in the nidopallium caudolaterale, often considered the equivalent of the mammalian prefrontal cortex. This was done while the animals performed a colour-change detection working memory task (the birds were required to determine if, following a memory delay, one out of three colored squares had changed its colour). Volitional attentional control was manipulated by comparing pre-cue trials, in which a cue indicated the only relevant location where a change could occur, and no-cue trials. This approach allowed to investigate how endogenous attentional control is performed by the pallium of birds, whose nuclear architecture is remarkably different from the laminar organization of mammalian cortices. Given the scarcity of studies on the neural bases of top-down attention control in non-mammalian species, this study is highly relevant for a wide community of scholars. Overall, this work starts to shed light on how the sophisticated processes required for this form of executive attentional control can be implemented in a non-laminar cortex homologue and on the evolutionary history of higher cognition.

Overall, the study is innovative, relevant and well-conducted, the methods are rigorous and the results are informative. I have, however, some minor changes to suggest, mostly to improve clarity. Please also keep in mind that formats in which the Results precede the Methods can be especially heavy for the readers.

Lines 67-93 here it is not very clear in which kind of trials (pre-cue or no-cue) did the Authors expect to observe volitional attentional modulation. What was the crucial comparison or the crucial type of trial/analysis for this study? This should be spelled out explicitly at this point of the text.

For instance, text says "[in no-cue trials] the birds made use of implicit strategies resulting in endogenous attentional selection. This allowed us to compare neuronal signals between conditions when a stimulus was attended, or when it was not". This suggests that endogenous attentional orienting should happen during no-cue trials, while pre-cue trials could represent a control condition in which attention is oriented exogenously. If this is the case, however, why most of the analyses reported at the beginning of result section seem to be restricted to the pre-cue trials? I would expect the crucial comparison/result to be presented first, ideally after the reader has been "cued" for it. Otherwise, the reader gets distracted by other aspects of these rich results and can be easily confused about which point the Authors are trying to make.

Some of these aspects are clarified at lines 341 and following, when the Authors nicely explain the logic and the approach that they followed a posteriori, given what the performance of the birds revealed on preferred locations. It would be great to have an equivalent paragraph at the end of the introduction, explaining the initial (a-priori) hypotheses that were behind the study design.

Lines 110-115: "Therefore, we tested if neuronal activity was different based on the factors:

location (presence of the pre-cue at either the top, middle, or bottom location), and color (color 1 or color 2 at any location, irrespective of the other colors on screen), or both.” Here, and in some parts of the following result section, I do not understand whether this analysis was run only for pre-cue trials. This should be spelled out clearly (and, if no-cue trials were also included, the description of the first factor should be changed to reflect this).

Lines 193-194: “When training bin and testing bin were the same or temporally close, the classification performance exceeded chance levels”. What are the implications of this fact (limitation to the case when the bins were temporally close) for the possibility to decode the relevant location based on the population activity?

At lines 523-525, when describing the putative correspondence between the entopallium and mammalian extrastriate cortexes, it would be worth considering also the evidence on the role of entopallium in pattern discrimination in zebra finches.

Version 1:

Reviewer comments:

Reviewer #1

(Remarks to the Author)

My comments have been addressed; I better understand several of the analyses and additional text on training, analysis of preference, and classifier results are useful. It is an interesting and rigorous study!

Reading through the manuscript and rebuttal letter, I do think a brief mention of the sample size and variability between crows in the discussion would be appropriate.

Reviewer #2

(Remarks to the Author)

The Authors have satisfactorily addressed all my comments.

**Reviewer #1 (Remarks to the Author):**

Neuronal correlates of endogenous selective attention in the endbrain of crows

**Summary**

The authors investigate an important research question, the neuronal correlates of attention in
a diverse animal model – the crow. Hahn and Fongaro et al., trained 2 crows on a visual change
detection task where 3 differently-colored squares were presented top, middle, and bottom
during the sample phase. After a delay in which the screen clears, the 3 squares reappear
where one of the squares with 50% probability could have changed color and crows have to
indicate whether or not this change occurred. Critically, this paradigm was applied to two
different trial types. In the first, a pre-cue (white circle) indicated the position in which the
square would change color if it did so and thus crows only had to attend to the colored square
at one spatial location. In the second no-cue condition, crows should have attended to color of
the squares at all three spatial locations. Crows were able to perform the task well in the pre-
cue trials and recordings in the nidopallium caudolaterale (NCL) revealed enhancement of
color information at the cued location. Interestingly, crows appear to leverage the structure of
the task in the no-cue condition and instead of attending to all 3 squares with some difficulty
given working memory constraints, picked one to attend to during the trial. Crow 1 changed
preferences and crow 2 preferred the top square. Authors were able to capitalize on this
behavior to examine differences in neural coding when squares were externally cued and when
they were internally preferred by the crows. The project is well-motivated, the analyses are
thorough, and I have a few comments I would like addressed.

Thank you for your review, we have addressed your comments and edited the manuscript
accordingly (line references indicate the location of changes in the manuscript).

**Major Comments:**

It is unclear to me in the no-cue trials, why crows have such a high performance in reporting
no change. In the discussion (Lines 401-403), the authors write ‘memorizing only one instead
of three colors allowed reporting a change at this location with very high performance (1/6
conditions) and to successfully report all trials without change (3/6). Why is it easier to detect
if no change had occurred if crows were attending to only one stimulus leading to ‘near perfect
correct rejection rates)?

We clarified this (lines 435 f.).

In short, the animal would perform nearly perfect for the one memorized location, reporting a
change correctly in 1/6 conditions (three locations * [change + no-change]). In trials without a
change the animal would report 'no change' by default (and thus 'near perfect' on correct
rejections), that is unless the memorized location changed, resulting in 3/6 correct responses.
Therefore, it would only give an incorrect response if one of the two unattended locations
changed (2/6).

Related to the above point, in the methods (Lines 646-648), 'If a color had changed, the birds
were trained to peck any of the original three squares to indicate the change. If no color had
changed, they were trained to peck at the square in the center.' If crows could peck at any of
the three squares, did they tend to peck at the square in which the change occurred or at the
square that they were presumably attending to?

We checked the pecking locations in correct change trials and observed that the animals
virtually always pecked on the cued locations (i.e., where the change occurred, and where, in
no-cue trials, we presume their attention was directed towards). Because there were
essentially no false alarms in pre-cue trials (pecking at a color that had not changed) we cannot
quantify if they pecked at a presumably attended location in error.

**We added a sentence mentioning where the birds pecked in the methods and in the**
**results sections (lines 120 f., 692 f.).**

In the section, Neuronal activity in pre-cue trials predicted location preference in no-cue trials,
the lack of a clear location preference of bird 1 was reflected in a lack of clear cross-trial
classification. However, this appeared to be from throughout a session. In a previous page
(Line 301), authors calculated a 'preferred location of the block' for bird 1. If analyses were
done on a block level for this bird, would there be change in decoding performance when the
data is not organized by spatial location but by preference? This would further support the
discussion that the same attentional processing is occurring in both trial types (Line 434-439)
which is currently focused on bird 2.

This is a good suggestion, and we initially tried to implement it on our analysis. However, due
to trial count limitations in our dataset, we cannot confirm block preferences of bird 1 in such a
way. Because bird 1 regularly shifted its preference between the different locations, the number
of correct trials for a given location is too low to support analysis of activity of the pseudo-
simultaneous neuronal population (see figure below). Neurons recorded in sessions without
sufficient trial numbers for a specific location would have to be removed from the pseudo-
simultaneous population, as these neurons cannot provide information about the specific trial
type.

 Figure R1: Hit rate of bird 1 in no-cue trials across sessions, for each target location (Fig. S2 B in the
 supplementary materials). Note the pattern in the right-hand plot that sessions commonly have low hit
 rates for two of the locations and a fair hit rate for one location. This pattern creates an issue when trying
 to analyze specific no-cue trial types/blocks using a pseudo-simultaneous neuronal population: neurons
 from sessions where a specific location had very few or even no correct trials must be excluded from
 the population, this would result in an underpowered analysis.

To get around these limitations we instead used all available correct no-cue trials of bird 1
 (across sessions, and irrespective of change location) to maximize the trial count and thus the
 neuronal population we use for classification, which gives us a comparable approach between
 birds, at the cost of precision, with respect to location preferences found in individual blocks
 within sessions.

We recognize that the low trial numbers are due to methodological shortcomings in the
 planning of trial type distribution. Ideally, we would have ensured that both birds produce an
 equal and consistent number of correct no-cue change trials across all three locations in all
 sessions. Given the data as they are we cannot meaningfully improve the strength of support
 for our results, we do, however, believe that our interpretation remains valid.

**We updated the discussion to indicate the limitations and point out that consistent trial**
 **numbers would be required to further confirm and expand our results (lines 469 f.).**

 The title and text reference 'endogenous attention' and the authors write that 'Exogenous
 attention is commonly defined as involuntary, caused by an unreliable and uninformative
 stimulus, of short durations [...]. In contrast, we used a fully reliable pre-cue that was
 predicteing the onset of the only relevant target color.' However, the pre-cue was present at
 the same spatial location as the square to be attended to. Could the authors discuss what
 might occur if the pre-cue was centrally located instead?

**We added a brief discussion on this in the section about 'Overt stimulus selection and**
 **eye movement' (lines 511 f.).**

We expect that, given enough training, the animals would have learned to use a symbolic or
more centralized cue to the same effect as observed in our protocol. As we discuss in detail,
the spatial co-occurrence of cue and target is not taking away from the endogenous nature of
the attention triggered by the cue, as long as the cue is reliable and not temporally
simultaneous with the target. The kind of cuing we employed is also commonly used to probe
endogenous attention in studies involving both human and non-human primates. The more
critical aspect is the possible role of eye movement and gaze fixation, as we also discuss in
detail.

There is little text on the training procedures of how the crows learned this task from a previous
study (Fongaro & Rose, 2020). Could you elaborate on different training stages to shape the
final behavior? Are there differences in the amount of training between crows?

**We added a short description of the training stages to the methods section (lines 707**
**f.).**

Overall differences in training amount between crows are very difficult to diagnose because
the animals had previously been participating in both the cited behavioral study by Fongaro &
Rose, and prior to that been trained in a similar study investigating working memory capacity
(Balakhonov & Rose, 2017). Thus, the training history is extensive, and training always formally
completed whenever each animal had reached the respectively preset criteria (of a certain
value of correct trials in all conditions). We do not believe that there were systematic
differences between animals other than the natural variation between animals.

Minor Comments:

The figures are a bit cluttered with black outlines around each subfigure.

**We removed the black outlines and tried to remove other cluttering elements.**

In Figure 1, the timeline of the task was initially a bit confusing with the different sizes of the
grey rectangles arranged before and behind each other.

**We simplified the time axis of the figure.**

In Figure 2, 'Purple shading indicates periods when spike-density between different cue
locations was significantly different'. It appears in C that there is purple shading even before
the onset of the cue?

**We have added a short note to the figure caption to alert the reader to the following:**

Spiking difference between the different conditions only appears for the second half of the cue
period (see spike raster). The purple shading indicates the range of significant bins. Data is
binned in sections of 200 ms, advanced in steps of 20 ms, therefore, for example, bins starting
at -100 ms relative to cue onset already contain spikes happening during the first 100 ms of
the cue period, and so forth. The PSTH therefore contains a systematic visualization artifact
that makes neuronal responses appear earlier than they do in the spike raster that has much
higher temporal resolution. Irrespective of binning method (i.e. temporal smoothing looking
forward, backward, or both), such systematic temporal offsets always appear in a PSTH.

We think both forms of visualization (spike raster and PSTH) are helpful and complimentary.
Timing of neuronal responses relative to discrete events should not be read out from PSTH
but from the spike raster instead. The PSTH only serves to make the spike density across trials
easier to read out from the visualization.

In Figure 2, can the authors speculate on why in Bird 1, there is no significant cluster for
decoding cue location in the pre-cue period? And more generally, why the temporal
consistency between phases seems limited (Line 208), which I would not expected given that
'endogenous attention [is] sustained (Line 45)'.

**We edited the section on the classifier results to provide more details regarding your**
**questions (lines from 211 f.).**

We generally interpret the data to reflect the location of directed endogenous attention, not as
the pre-cue location per se (even though in the pre-cue trials they obviously coincide). In no-
cue trials bird 1 did not have an obvious location of directed attention but may have decided
'on the spot' where to pay attention to (unlike bird 2 that always directs attention to the top
location). We think this is reflected in the classifier results of the pre-cue location as seen in
Fig. 2C. The neuronal population of bird 1 may have been less stable (and thus less predictive
of cue-location, resulting in lower classifier performance) prior to the sample onset when
attention was then directed to the relevant location (increasing classifier performance). In
contrast bird 2 was hyper focused on the top location (to a degree where it seems like it even
missed the pre-cue in some trials, as seen from the more spread-out performance in Fig. 1 C),
which might have stabilized neuronal activity and subsequently classifier performance.

Regarding temporal consistency and sustained endogenous attention: While the phenomenon
of attention appears as sustained, the underlying neuronal population may not actually produce
a sustained form of activation, observable as spiking or classification readout depending on
spiking. Instead, different parts of the neuronal population may be active or silent at different
167 times, analogous to such conceptualizations of working memory, which is deeply entwined
with attention. We therefore think the population of bird 1 might undergo a form of silent activity

during the delay to maintain information and is read out and 'reactivated' in the comparison
phase. In contrast, the population of bird 2 may have a more continuous appearance because
the read out of the preferred top location is more dominant.

In Figure 2, why does there appear to be a division in the cue information present (e.g. classifier
performance) between the beginning and the end of the delay phase in both birds (E and F)?

**We edited the section on the classifier results to provide more details regarding this**
**question (lines from 211 f.).**

Cue location information is important during both the sample and comparison phase to identify
the relevant sample color and if it changed, but less relevant during the middle part of the
delay, where color information might be more relevant (see for example the neuron in Fig. 4
B). Therefore, it is conceivable that neuronal population information about cue location
(indicated by classifier performance) waxes and wanes with peaks in sample and comparison
and a trough in the middle of the delay. This description fits best to bird 1 and only in limited
fashion to bird 2, who might have had a different overall neuronal pattern due to its distinct
strategy as described above concerning the previous question.

Do crows have a preference for the color they would like to attend to out of the 15 possible
colors when examined together? Is that what is calculated for color in 2B or is that just between
the pairs of colors (Figure 3)?

In 2 B we show the percentage of neurons that exclusively had a significant response for a
given color pair in a session (as in Fig. 3), we did not observe any color preference, and
furthermore did not systematically test for it either. Because there were always three colors
displayed simultaneously, individual colors can never fully be analyzed in isolation, only as part
of the presented three color configurations. Encoding of color has been reported in pigeon NCL
by Johnston et al. (2017), which we mention in the discussion, but we do not think that it is
within the scope of our study to test for this directly, because it was not the exact color identity
that mattered in our task, only the distinct change of colors.

There were 850 trials in the session and sessions were broken into blocks of 5 for the
preference calculation. How was this block size chosen? Are the preferences robust if the block
size is adjusted to be smaller or larger?

**We realize that there were insufficient details on how we analyzed the preference of the**
**birds, please see below for clarifications that we also added to a new methods section**
**'Behavioral analysis of preference' (lines 748 f.).**

First, depending on how well the bird behaved regarding the camera head tracking, different
 amounts of trials per session (out of the 850) were completed and subsequently analyzed. On
 average 456.12 trials (range from 343 and 573 trials) per session were analyzed for bird 1, and
 402.2 trials (range from 322 to 536 trials) for bird 2.

We chose a block size of 20 % of completed trials of a session to estimate preference. We
 think 20 % gives a reasonable number of trials for analysis, while keeping the result intuitive
 to understand as 'fifths of a session'. The preferences were robust for a wide range of chosen
 block sizes (see below). For all block sizes bird 1 showed a small preference for the bottom
 location, over the top location, and a clearly least preferred middle location. Bird 2 always
 overwhelmingly preferred the top location.

Fig. R2: Preference estimates for bird 1 (left) and bird 2 (right). Same methodology as in Fig. 5 C/D
 (where only values for $n = 5$ are shown), with different quantile sizes of the session ranging from thirds
 to tenths. The qualitative results were not affected by the size of chosen quantiles.

To further clarify: We report preference as the fraction of all quantiles, across all sessions that
 have the respective location preference out of all quantiles of all sessions. For example: bird 2
 performed in a total of 12 sessions. We performed analysis on five blocks per session (i.e., on
 non-overlapping blocks, each consisting of 20 % of completed trials), yielding a preferred
 location for each block (60 in total). Out of those 60 blocks, bird 2 preferred the top location 59
 223 times (98.33 %), and the bottom location 1 time (1.66 %; see Fig. 5 C and Fig. R1 at $n = 5$).

**Reviewer #2 (Remarks to the Author):**

This paper investigates the neural correlates of endogenous attentional orienting in birds, using
crows as a model species. Extracellular single unit recordings were performed in the
nidopallium caudolaterale, often considered the equivalent of the mammalian prefrontal cortex.
This was done while the animals performed a colour-change detection working memory task
(the birds were required to determine if, following a memory delay, one out of three colored
squares had changed its colour). Volitional attentional control was manipulated by comparing
pre-cue trials, in which a cue indicated the only relevant location where a change could occur,
and no-cue trials. This approach allowed to investigate how endogenous attentional control is
performed by the pallium of birds, whose nuclear architecture is remarkably different from the
laminar organization of mammalian cortexes. Given the scarcity of studies on the neural bases
of top-down attention control in non-mammalian species, this study is highly relevant for a wide
community of scholars. Overall, this works starts to shed light on how the sophisticated
processes required for this form of executive attentional control can be implemented in a non-
laminar cortex homologue and on the evolutionary history of higher cognition.

Overall, the study is innovative, relevant and well-conducted, the methods are rigorous and
the results are informative. I have, however, some minor changes to suggest, mostly to
improve clarity. Please also keep in mind that formats in which the Results precede the
Methods can be especially heavy for the readers.

Thank you for your review, we have addressed your comments and edited the manuscript
accordingly (line references indicate the location of changes in the manuscript).

Lines 67-93 here it is not very clear in which kind of trials (pre-cue or no-cue) did the Authors
expect to observe volitional attentional modulation. What was the crucial comparison or the
crucial type of trial/analysis for this study? This should be spelled out explicitly at this point of
the text.

For instance, text says “[in no-cue trials] the birds made use of implicit strategies resulting in
endogenous attentional selection. This allowed us to compare neuronal signals between
conditions when a stimulus was attended, or when it was not”. This suggests that endogenous
attentional orienting should happen during no-cue trials, while pre-cue trials could represent a
control condition in which attention is oriented exogenously. If this is the case, however, why
most of the analyses reported at the beginning of result section seem to be restricted to the
pre-cue trials? I would expect the crucial comparison/result to be presented first, ideally after
the reader has been “cued” for it. Otherwise, the reader gets distracted by other aspects of

these rich results and can be easily confused about which point the Authors are trying to make.

Some of these aspects are clarified at lines 341 and following, when the Authors nicely explain
the logic and the approach that they followed a posteriori, given what the performance of the
birds revealed on preferred locations. It would be great to have an equivalent paragraph at the
end of the introduction, explaining the initial (a-priori) hypotheses that were behind the study
design.

**We edited the last paragraph of the introduction and first paragraph of the results**
**outlining our original reasoning behind the study approach to help the reader follow our**
**train of thought (lines 75 f., 102 f.).**

We reasoned that reporting the results of the pre-cue trials first would allow for a more intuitive
understanding of the overall results. Analysis on the pre-cue trials is relatively straightforward
as it is limited to the cue location and the colors at the cue location. Furthermore, we argue
that the main effect of endogenous attention can already be read out from the pre-cue trials.
Crucially, the analysis of the no-cue trials adds to that by providing evidence of the same
mechanism being present even without the external cuing.

Lines 110-115: "Therefore, we tested if neuronal activity was different based on the factors:
location (presence of the pre-cue at either the top, middle, or bottom location), and color
(color 1 or color 2 at any location, irrespective of the other colors on screen), or both." Here,
and in some parts of the following result section, I do not understand whether this analysis was
run only for pre-cue trials. This should be spelled out clearly (and, if no-cue trials were also
included, the description of the first factor should be changed to reflect this).

**We clarified which trial types were used (lines 102 f. and line 128 f.).**

These analyses were conducted exclusively for pre-cue trials. Pre-cue location (factor 1)
cannot be tested if there was no cue (and we think testing cue locations vs. absence of the
cue would not add a meaningful interpretation). Furthermore, having the cue located at a
specific location allowed us to focus on the color pair shown at that location.

Lines 193-194: "When training bin and testing bin were the same or temporally close, the
classification performance exceeded chance levels". What are the implications of this fact
(limitation to the case when the bins were temporally close) for the possibility to decode the
relevant location based on the population activity?

**We edited the section on the classifier results to provide more details regarding your**
**question and to clarify some points raised by reviewer 1 (lines 211 f.).**

While the phenomenon of attention appears as sustained, the underlying neuronal population
may not actually produce a sustained form of activation. Instead, information (successful
classification) may only be present in a localized fashion, both temporally and in different parts
of the neuronal population, which may be active or silent at different times, analogous to such
conceptualizations in models of working memory, which is deeply entwined with attention.

For example, the neuronal population of bird 1 seemingly 'loses' cue location information in
the delay, before returning to a high level of information in the comparison phase, which mirrors
the initial level of information during the sample phase. So, there is actually a very high
temporal stability between sample and comparison phase, it's just that the delay phase
between them is not included in the same neuronal activity pattern. In contrast, the population
of bird 2 may have a more continuous appearance because the read out of the preferred top
location is more dominant, and possibly relying on a more distributed neuronal population that
ends up tiling the delay phase.

At lines 523-525, when describing the putative correspondence between the entopallium and
mammalian extrastriate cortexes, it would be worth considering also the evidence on the role
of entopallium in pattern discrimination in zebrafinches.

**We added a mention of the work done in zebrafinches and pigeons in the discussion.**

**Lines 576 f.:** The ENT may be of particular interest as it is a hub for visual processing that
may be functionally comparable to parts of extrastriate cortex of mammals (Clark & Colombo,
2020), playing a crucial role in pattern discrimination in pigeons (**Hodos & Karten, 1970,**
**Watanabe, 1991**) and zebrafinches (**Watanabe et al., 2008; 2011**).